# Incidence and Risk Factors for Glucose Disturbances in Premature Infants

**DOI:** 10.3390/medicina58091295

**Published:** 2022-09-16

**Authors:** Ivona Butorac Ahel, Kristina Lah Tomulić, Inge Vlašić Cicvarić, Marta Žuvić, Kristina Baraba Dekanić, Silvije Šegulja, Iva Bilić Čače

**Affiliations:** 1Department of Pediatrics, Clinical Hospital Center Rijeka, 51000 Rijeka, Croatia; 2Faculty of Medicine, University of Rijeka, 51000 Rijeka, Croatia; 3Pediatric Intensive Care Unit, Clinical Hospital Center Rijeka, 51000 Rijeka, Croatia; 4Department of Clinical, Health and Organizational Psychology, Clinical Hospital Center Rijeka, Krešimirova 42, 51000 Rijeka, Croatia; 5Department of Biotechnology, University of Rijeka, Radmile Matejčić 2, 51000, Rijeka, Croatia; 6Department of Clinical Science, Faculty of Health Studies, University of Rijeka, Viktora Cara Emina 5, 51000 Rijeka, Croatia; 7Neonatal Intensive Care Unit, Department of Pediatrics, Clinical Hospital Center Rijeka, Krešimirova 42, 51000 Rijeka, Croatia

**Keywords:** preterm infant, hypoglycemia, hyperglycemia, glucose variability

## Abstract

Background and Objectives: There are limited data regarding the incidence and risk factors for hypoglycemia, hyperglycemia, and unstable glycemia in preterm infants. The aim of the present study was to determine the incidence and risk factors associated with neonatal hypoglycemia, hyperglycemia, and unstable glycemia in preterm infants during the first seven days of life. Materials and Methods: This prospective study included preterm infants <37 weeks of gestation, admitted to the Neonatal Intensive Care Unit between January 2018 and December 2020. Based on blood glucose levels in the first week of life, infants were divided into the following four groups: normoglycemic, hypoglycemic, hyperglycemic, and unstable. Blood glucose levels were measured from capillary blood at the 1st, 3rd, 6th, and 12th hour of life during the first 24 h, and at least once a day from days 2 to 7, prefeed. Results: Of 445 enrolled infants, 20.7% (92/445) were categorized as hypoglycemic, 9.9% (44/445) as hyperglycemic, and 2.9% (13/445) as unstable, respectively. Hypoglycemia was most commonly observed among infants ≥34 weeks (27.9%), and hyperglycemia was most common among preterm infants <28 weeks (50%). Female gender increased the chances of developing hypoglycemia by three times. The decrease in gestational age by one week increased the chance of developing hyperglycemia by 1.9 times. Sepsis increased the chance of developing hyperglycemia seven times, respiratory distress syndrome five times, and mechanical ventilation three times, respectively. Conclusions: Glucose disturbances in the early neonatal period in preterm infants are common and mostly asymptomatic. Therefore, careful blood glucose level monitoring is required in those infants, especially in late preterm infants, in order to prevent possible neurological complications.

## 1. Introduction

Glucose homeostasis presents a major challenge for preterm infants, especially during the first weeks of life. Preterm infants are prone to blood glucose level fluctuations after birth due to limited supplies of energy sources for carbohydrate metabolism and immaturity of organs involved in the regulation of energy metabolism [1]. There are many unanswered questions regarding the definition, thresholds for intervention, and management strategies of both hyperglycemia and hypoglycemia. There are also many disagreements about the effect of glucose imbalance on neurodevelopmental outcomes [2,3,4].

Hyperglycemia is one of the most common metabolic conditions in extremely low birth weight infants (ELBW). It has been associated with significant morbidity and mortality, including intraventricular hemorrhage, retinopathy of prematurity, necrotizing enterocolitis, and abnormal neurologic examination at two years of age [5,6,7,8]. Several risk factors contribute to the development of hyperglycemia in preterm infants, including gestational age, low birth weight, stress, excessive glucose administration, parenteral nutrition, and the administration of some drugs, such as corticosteroids, phenytoin, or theophylline [9,10,11]. Hypoglycemia is a very common, usually transient, metabolic disturbance in newborn infants as a result of an adjustment to the extrauterine environment. Hypoglycemia occurs in 5–10% of otherwise healthy infants, and the incidence increases in preterm infants [12]. Symptomatic and recurrent hypoglycemia has been associated with structural brain abnormalities, impaired neurodevelopment, and poor school performance [13,14]. The fact that hypoglycemia is often asymptomatic presents another challenge regarding its effect on brain damage. Antenatal steroid administration, maternal hypertension, maternal diabetes, and being born as small (SGA) or large (LGA) for gestational age have been identified as risk factors for developing hypoglycemia [12,15,16]. Although glucose variability in preterm infants has been the subject of many discussions recently, data about the incidence, risk factors, and the impact of glucose variability in those patients are insufficient and scarce [17].

The aim of the present study was to determine the incidence and risk factors associated with hypoglycemia, hyperglycemia, and unstable glycemia in preterm infants during the first seven days of life.

## 2. Materials and Methods

### 2.1. Study Group

This single-center prospective study included all preterm infants <37 weeks of gestation born between January 2018 and December 2020 and admitted to the Neonatal Intensive Care Unit (NICU) in the Clinical Hospital Center Rijeka. The exclusion criteria were as follows: admission to the NICU after 24 h of age, death or discharge before postnatal day 7, and major congenital malformation, including chromosome abnormality. During the study period, there were 482 preterm infants admitted to the NICU. Parents of 16 infants refused to participate, and 21 were excluded from the study due to death before postnatal day 7 (*n* = 6), severe congenital anomalies which needed surgical procedure in the first 7 days of life (*n* = 13), and chromosome abnormalities (*n* = 2). Finally, a total of 445 infants were enrolled in the study based on the inclusion/exclusion criteria.

Based on measured blood glucose levels during the first week of life, infants were classified as follows:-Normoglycemic (a single blood glucose level of 2.1–2.5 mmol/L (38–45 mg/dL) or a single blood glucose level of 8.6–10.0 mmol/L (155–180 mg/dL), with all other measurements 2.6–8.5 mmol/L (47–153 mg/dL));-Hypoglycemic (blood glucose level of ≤2.5 mmol/L (45 mg/dL) on ≥2 measurements >1 h apart, or any blood glucose level ≤ 2.0 mmol/L (36 mg/dL));-Hyperglycemic (blood glucose level ≥ 8.6 mmol/L (155 mg/dL) on ≥2 measurements >1 h apart, or any blood glucose level ≥ 10.1 mmol/L (182 mg/dL));-Unstable (≥1 blood glucose level ≤ 2.5 mmol/L (45 mg/dL) and ≥1 blood glucose level ≥ 8.6 mmol/L (155 mg/dL)) [4].

According to gestational age (GA), preterm infants were divided into the following four groups: GA < 28 weeks, GA between 28 and 31 weeks, GA between 32 and 33 weeks, and GA ≥ 34 weeks. Infants whose birth weights were below the 10th percentile for gestational age and sex were categorized as small for gestational age (SGA), and those above the 90th percentile were classified as large for gestational age (LGA) according to the Fenton preterm growth chart [18].

Demographic and perinatal characteristics of all infants were collected from the charts, and these included birth weight, gender, gestational age, multiple pregnancies, mode of delivery, prenatal steroid administration, maternal hypertension, and maternal diabetes. Neonatal outcomes, including early sepsis, necrotizing enterocolitis, mechanical ventilation, and respiratory distress syndrome, were recorded.

### 2.2. Methods

Blood glucose levels were measured from capillary blood at the 1st, 3rd, 6th, and 12th hour of life during the first 24 h, and at least once a day from days 2 to 7 prefeed. Capillary blood samples were taken after the heels of the babies were heated. Blood glucose levels were determined using the gas analyzer RAPIDPoint 500. On the first day of life, all infants’ initial glucose infusion rates (GIR) were 4.5 mg/kg/min, and these were achieved either by intravenous infusion of parenteral nutrition, via enteral feeds alone, or in combination with intravenous 10% glucose infusion. The GIR was titrated and increased by 0.5 to 1 mg/kg/min daily up to 10 to 12 mg/kg/min, to maintain blood glucose levels within the target range of 2.5 to 8.6 mmol/L (45 to 155 mg/dL). The minimal GIR could not be lower than 4 mg/kg/min. The parenteral nutrition was increased daily to reach the nutritional recommendations of a total intake of 3.5–4 g/kg/day of proteins and 2.5–3 g/kg/day of lipids, with an average energy intake of 120 kcal/kg/day. Enteral feeding was started as soon as possible after birth, using breast milk as the first choice. In case of symptomatic hypoglycemia or glucose level ≤ 2.2 mmol/L (40 mg/dL), bolus infusion of 10% dextrose—2 mL/kg was administered over 10 min, and, if necessary, GIRs were increased to reach the desired levels of blood glucose. In case of glucose level > 2.2 mmol/L (40 mg/dL) without obvious symptoms of hypoglycemia, the infant ≥34 GA received oral feeding. In cases where the overall condition of the infant did not allow for oral feeding, they were fed by a gastric tube. Infants <34 GA without symptoms of hypoglycemia and glucose level > 2.2 mmol/L (40 mg/dL) received 40% dextrose buccally. After any kind of intervention for hypoglycemia, blood glucose levels were measured again after 20 min. The first-line approach for hyperglycemia was to reduce the rate of GIR by 1–2 mg/kg/min. If glucose reduction failed to lower glycemia and blood glucose levels were >14 mmol/L (252 mg/dL), the second line approach was the introduction of an intravenous insulin infusion. Insulin treatment was started at 0.05 U/kg/h and was titrated by capillary measurements to maintain a glycemia of 4–10 mmol/L (72–180 mg/dL).

The study has been registered on Clinicaltrials.gov (No. NCT 03530189).

### 2.3. Ethical Statement

The study followed the guidelines of the Declaration of Helsinki, and the procedures were approved by the Ethics Committee of the Clinical Hospital Center Rijeka (No. 2170-29-02/15-17-2). All the parents of the participants provided their written informed consent to participate in this study.

### 2.4. Statistical Analyses

The collected data were statistically analyzed using the data analysis software systems TIBCO Statistica 14.0.0 (2020) and MedCalc 17.9.2 (2017). Categorical and counting variables were presented as frequencies or percentages and analyzed using Pearson’s chi-square test. The distribution of continuous variables was tested for normality using the Shapiro–Wilk test, and comparisons between glycemic categories (normoglycemia, hyperglycemia, hypoglycemia, and unstable) were made using a one-way analysis of variance (ANOVA test) with a post hoc Scheffé test. Predictors of glycemic status were analyzed using univariate and multivariate logistic regression modeling, with the predictive power of the model assessed using Nagelkerke R2 and AUC. Individual predictors were additionally assessed using ROC analysis, with AUC > 0.80 indicating good discrimination and AUC > 0.90 indicating excellent discrimination. Values of *p* < 0.05 were considered to indicate statistical significance.

## 3. Results

### 3.1. Characteristics of the Studied Group

The baseline characteristics of the studied group of preterm infants are presented in Table 1. The median GA of the study population was 32 ± 3 weeks. Of 445 enrolled infants, 66.5% (296/445) were categorized as normoglycemic, 20.7% (92/445) as hypoglycemic, 9.9% (44/445) as hyperglycemic, and 2.9% (13/445) as unstable. The hyperglycemic and the unstable group of infants were of significantly lower gestational age (ANOVA, F = 61.9, *p* < 0.001), lower birth weight (ANOVA, F = 39.6, *p* < 0.001), lower 1-(ANOVA, F = 29.5, *p* < 0.001) and (ANOVA, F = 38.5, *p* < 0.001), higher CRIB II scores (ANOVA, F = 31.7, *p* < 0.001), 5-minute Apgar score, and had longer hospital stays (ANOVA, F = 73.8, *p* < 0.001) than those in both the normoglycemic and hypoglycemic group. Infants in the unstable group had higher CRIB II scores than infants in all other groups. Hypoglycemic episodes were the most common in preterm infants ≥34 weeks; 51 of 183 infants in this subgroup developed hypoglycemia (27.9%). Symptomatic hypoglycemia was seen in seven infants (1.6%). Observed hypoglycemia symptoms were jitteriness in four infants, poor feeding in two infants, and lethargy in 1 infant. Recurrent episodes of hypoglycemia were recorded in six infants (1.3%), and all of them were asymptomatic. They did not occur in any infant after the third day of life, and we did not perform critical sampling in order to investigate an underlying cause. Hyperglycemic episodes were most common in preterm infants <28 weeks; 23 of the 46 (50%) developed hyperglycemia. A total of 17 preterm infants (38.6%) in the hyperglycemic group received insulin. The lowest mean blood glucose level was recorded in the hypoglycemic category 2 h after birth (2.7 ± 1.1 mmol/L, 49 ± 20 mg/dL). The highest mean blood glucose level was recorded in the hyperglycemic category on the second day of life (11.0 ± 6.0 mmol/L, 198 ± 108 mg/dL).

### 3.2. Risk Factors for the Development of Hypoglycemia

The results of the univariate analysis showed that being female increased the chances of developing hypoglycemia. In the multivariate analysis, gender, gestational age, and CRIB II remained risk factors. Female infants had a three-times higher risk of developing hypoglycemia. This model had a weak predictive power (Table 2). The ROC analysis showed that blood glucose levels ≤ 2.6 mmol/L (47 mg/dL) measured immediately after birth and in the third hour of life were an excellent predictor of hypoglycemia with high specificity. Gestational age, CRIB II, and blood glucose level on the seventh day of life had no predictive value for developing hypoglycemia (low AUC).

### 3.3. Risk Factors for the Development of Hyperglycemia

In the univariate analysis, gestational age, blood pH at birth, sepsis, RDS, and mechanical ventilation were the best correlates with the hyperglycemic category. In the multivariate analysis, gestational age, sepsis, RDS, and mechanical ventilation remained highly significant predictors. This multivariate regression model showed high clinical significance and good predictive power (2 log likelihood = 182.22; *p* < 0.001, R2 (Nagelkerke) = 0.487; AUC = 0.874 ± 0.028, *p* < 0.001) (Table 3). A decrease in GA by one week increased the chance of developing hyperglycemia 1.9 times. Sepsis increased the chance of developing hyperglycemia seven times, RDS five times, and mechanical ventilation three times. Blood pH taken from capillary blood immediately after birth was a significant but not good predictor of hyperglycemia (lower AUC).

### 3.4. Risk Factors for the Development of Unstable Glycemia

In the univariable analysis, being female and having lower base excess at birth were identified as being significant predictors of unstable glycemia. In the multivariate regression analysis, there was a significant association between the risk of unstable glycemia and female gender, CRIB II score, and lower base excess. The number of infants with unstable glycemia was small (N = 13) and, thus, the significance of these results is questionable.

## 4. Discussion

In the present study, the incidence of hypoglycemia in preterm infants was 21%. In previous reports, the incidence of hypoglycemia in this population ranged between 9% and 42%. This wide range was attributed to the use of different definitions of hypoglycemia, the timing of blood glucose testing, and the feeding status [4,14,19,20,21,22,23,24]. In our study, most cases of hypoglycemic episodes occurred in the 1st and the 3rd hour of life (80%). The first 2 or 3 h of life are the periods when low blood glucose levels are usually transient and considered to be part of normal adaptation to postnatal life. This adaptation process is more difficult for preterm infants to accomplish than term infants, because they have low glycogen and fat stores with limited capacity for gluconeogenesis [25]. Hypoglycemic episodes most commonly affected infants born ≥34 weeks of gestation (28%) which was unexpected. They were also the largest group of preterm infants in our study (41%). Infants born ≥34 weeks of gestation, also called late preterm infants, represent the most numerous subgroup of preterm infants. Although they resemble term infants, they are immature metabolically and physiologically, and are consequently prone to multiple complications including hypoglycemia, feeding difficulties, hyperbilirubinemia, and respiratory distress [26,27]. Celik et al. reported that 4% of late preterm infants were admitted to the NICU due to hypoglycemia. The proportion of hypoglycemia in late preterm infants increased to 14.5% on follow-up [27]. Hosagasi et al. reported that the incidence of hypoglycemia among late preterm infants was 34% [28]. Teune et al. performed a systematic review and reported a hypoglycemia rate of 7.1% in late preterm infants [29]. In comparison with term infants, late preterm infants have a higher risk of developing hypoglycemia [30]. Our finding that late preterm infants had the most hypoglycemic episodes could be explained by the immaturity of the gastrointestinal tract and inadequate oral intake. In this group, a target GIR of 4.5 mg/kg/min was mainly attempted to be achieved by enteral feeding alone, or in combination with intravenous 10% glucose infusion. Meeting adequate nutritional needs in preterm infants presents a significant challenge due to immaturity of the digestive and absorptive processes and gastrointestinal motility [31]. Furthermore, oral feeding in late preterm infants has been challenging and inadequate due to immature oro-buccal coordination and swallowing mechanisms [32]. Due to the aforementioned issues, we cannot be sure that we achieved the target GIR. These results stress the importance of a careful postnatal follow-up of late preterm infants.

Previous studies have demonstrated that LGA or SGA infants, infants of diabetic mothers, and infants of hypertensive mothers are at increased risk of hypoglycemia [14,33,34,35,36]. These factors were not identified as risk factors for neonatal hypoglycemia in our population. Gender was not determined as a risk factor for developing hypoglycemia in previous reports [14,19]. In our study, female preterm infants were found to have a three-times higher risk of developing hypoglycemia. Our finding is, perhaps, a result of some undetected confounding factor. Future studies will be needed to confirm this finding.

The risk of hyperglycemia is inversely related to gestational age. Our study showed that the decrease in GA by one week increased the chance of developing hyperglycemia 1.9 times. Hyperglycemia most commonly affects ELBW infants in the first week of life. In previous reports, the incidence of hyperglycemia in ELBW in the neonatal period ranges from 38% to 85% [9,37,38]. The incidence of hyperglycemia in our study was 9.9%, which is significantly lower than in other similar studies [5,19]. This can be attributed to the fact that our sample includes all premature infants regardless of gestational age. If the ELBW infants are observed separately from the rest of our sample, the incidence of hyperglycemia is 51%, which is similar to previous studies. The pathogenesis of hyperglycemia in preterm infants is complex and multifactorial. It results from a combination of the inability to suppress glucose production despite glucose infusion, the lack of insulin-sensitive tissues, and relative insulin deficiency [11]. In addition, sepsis, necrotizing enterocolitis, and respiratory distress may lead to insulin resistance and secondary hyperglycemia. Our study showed that sepsis increases the chance of developing hyperglycemia seven times, RDS five times, and mechanical ventilation three times, respectively. Stressful events, including sepsis, might trigger hyperglycemia by increasing the secretion of pro-inflammatory mediators, which lead to insulin resistance, as well as altered insulin receptor signaling [39]. Additionally, counter-regulatory hormones and the consequent stimulation of glycogenolysis and gluconeogenesis play an important role in the pathogenesis of hyperglycemia.

Our results showed that preterm infants delivered by cesarean section were more likely to develop neonatal hyperglycemia (OR = 2.6, 95% CI: 1.3–5.5 (*p* < 0.001). Cesarean section was the most used mode of delivery among preterm infants in the hyperglycemic group. Infants in this group were the most preterm and had the lowest GA, showing that this association between hyperglycemia and cesarean section might not be causal. Previous studies reported cesarean section as a risk factor for hypoglycemia [40,41,42]. Further studies are needed to investigate and explain the connection of hypoglycemia and hyperglycemia to cesarean section.

Our study had several limitations. Firstly, this was a single-center study, and the sample size was limited. Hence, our findings need to be verified by multicenter and larger cohort studies. Secondly, we performed only intermittent blood glucose sampling and probably missed some abnormal values. Hyperglycemia and hypoglycemia can be prolonged and clinically silent but are associated with worse developmental outcomes in childhood. These events can be early identified early by using CGM. Additional studies are needed to determine the incidence of hypoglycemia and hyperglycemia by using CGM. Nevertheless, the prospective nature of our study is its strength.

## 5. Conclusions

Since hypo- and hyperglycemia in the early neonatal period in preterm infants are common and mostly asymptomatic, careful blood glucose level monitoring is required. Additional attention should be directed toward late preterm infants who comprise the majority of all preterm births and are often treated like term infants despite their risk of morbidity. Regular monitoring of glucose levels in preterm infants in the first days of life should enable early intervention and, thus, prevent possible neurological complications.

## Figures and Tables

**Table 1 medicina-58-01295-t001:** Baseline clinical and demographic characteristics of the studied group.

	Glycemic Category, *n (%)*				
	Normoglycemic 296 (66.5%)	Hypoglycemic 92 (20.7%)	Hyperglycemic 44 (9.9%)	Unstable 13 (2.9%)	Total 445
Infant data					
Female, *n* (%)	134 (45.3%)	49 (53.3%)	22 (50.0%)	4 (30.8%)	209 (46.9%)
Gestational age (weeks) **	33 ± 3	33 ± 2	28 ± 2	28 ± 3	32 ± 3
<28 weeks GA, *n* (%)	14 (4.7%)	1 (1.1%)	23 (52.3%)	8 (61.5%)	46 (10.3%)
28 to ≤31 weeks GA, *n* (%)	85 (28.7%)	19 (20.7%)	17 (38.6%)	2 (15.4%)	123 (27.6%)
32 to ≤33 GA weeks, *n* (%)	66 (22.3%)	21 (22.8%)	4 (9.1%)	2 (15.4%)	93 (20.9%)
≥34 GA weeks, *n* (%)	131 (44.3%)	51 (55.4%)	0 (0.0%)	1 (7.7%)	183 (41.1%)
Birth weight (g) **	1975 ± 598	1952 ± 572	1106 ± 445	1001 ± 419	1856 ± 646
SGA, *n* (%) LGA, *n* (%)	166 (56.1%)	61 (66.3%)	30 (68.2%)	11 (84.6%)	268 (60.2%)
	5 (1.7%)	1 (1.1%)	0	0	6 (1.3%)
Apgar score at 1 min **	8 ± 2	8 ± 2	5 ± 2	5 ± 2	8 ± 2
Apgar score at 5 min **	9 ± 1	9 ± 1	7 ± 2	6 ± 2	8 ± 2
CRIB II score **	4 ± 3	4 ± 3	8 ± 3	10 ± 3	5 ± 4
Length of hospital stay (days) **	20 ± 18	18 ± 15	73 ± 54	64 ± 44	26 ± 30
Blood gas at birth					
pH *	7.33 ± 0.07	7.32 ± 0.09	7.30 ± 0.08	7.28 ± 0.08	7.32 ± 0.08
pCCO_2_	5.8 ± 1.9	6.0 ± 1.6	6.3 ± 1.6	5.7 ± 1.3	5.9 ± 1.8
Base excess *	−3.1 ± 4.0	−3.7 ± 3.1	−3.3 ± 2.9	−6.2 ± 4.0	−3.3 ± 4.0
Maternal data					
Cesarean section, ** *n* (%)	167 (56.4%)	56 (60.9%)	34 (77.3%)	9 (69.2%)	266 (59.8%)
Maternal gestational or pre-existing diabetes mellitus, *n* (%)	48 (16.2%)	15 (16.3%)	5 (11.4%)	2 (15.4%)	70 (15.7%)
Pregnancy-induced hypertension, *n* (%)	38 (12.8%)	12 (13.0%)	11 (25.0%)	0	61 (13.7%)
Antenatal glucocorticoids, ** *n* (%) (administered to infants <34 GA)	160 (54.1%)	42 (45.7%)	43 (97.7%)	12 (92.3%)	257 (57.8%)
Postnatal complications					
Necrotizing enterocolitis, ** *n* (%)	10 (2.2%)	4 (1.35%)	2 (2.2%)	3 (6.82%)	1 (7.7%)
Sepsis ** *n* (%)	13 (4.4%)	2 (2.2%)	12 (27.3%)	3 (23.1%)	30 (6.7%)
Mechanical ventilation, ***n* (%)	83 (28.0%)	14 (15.2%)	33 (75.0%)	9 (69.2%)	139 (31.2%)
RDS ** *n* (%)	66 (14.8%)	35 (11.2%)	6 (6.52%)	21 (47.7%)	4 (3.8%)

Data are presented as mean ± standard deviation or number (%). Here, * *p* ≤ 0.05; ** *p* < 0.001. Abbreviations are as follows: GA—gestational age; SGA—small for gestational age; LGA—large for gestational age; CRIB—clinical risk index for babies; RDS—respiratory distress syndrome.

**Table 2 medicina-58-01295-t002:** Univariate and multivariate logistic regression for the prediction of hypoglycemia based on perinatal characteristics of preterm infants, mode of delivery, maternal data, and complications.

Variable	Univariate Analysis					Multivariate Analysis				
	Beta	*p*	OR	95% CI	*P*	Beta	*p*	OR	95% CI	*p*
Sex (female)	1.239	0.032	3.45	1.11–10.7	0.038	1.136	0.030	3.12	1.11–3.72	0.031
GA (weeks)	0.262	0.347	1.30	0.75–0.25	0.561	0.408	0.078	1.50	0.95–2.36	0.061
SGA	0.098	0.869	1.10	0.34–3.56	0.871	Eliminated				
LGA	−19.22	0.999	0.00			Eliminaated				
Vaginal delivery	0.121	0.830	1.13	0.37–3.39	0.852	Eliminated				
Apgar score at 1 min	−0.439	0.193	0.64	0.33–1.25	0.458	Eliminated				
Apgar score at 5 min	0.677	0.130	1.97	0.82–4.72	0.568	Eliminated				
CRIB II	0.223	0.193	1.25	0.89–1.75	0.462	0.227	0.096	1.25	0.96–1.64	0.062
pH	−4.126	0.283	0.02	0.00–29.9	0.856	Eliminated				
BE	0.027	0.606	1.03	0.93–1.14	0.269	Eliminated				
pCO_2_	−0.086	0.492	0.92	0.72–1.17	0.412	Eliminated				
Antenatal glucocorticoids	20.346	0.999	0.00			Eliminated				
Pregnancy-induced hypertension	−0.285	0.685	0.75	0.19–2.98	0.563	Eliminated				
Maternal diabetes	−0.635	0.462	0.53	0.09–2.88	0.696	Eliminated				
Sepsis	−0.729	0.523	0.48	0.05–4.53	0.795	Eliminated				
NEC	0.253	0.849	1.29	0.09–17.4	0.981	Eliminated				
RDS	−1.203	0.130	0.30	0.06–1.42	0.691	Eliminated				

Data are presented as odds ratios for developing hypoglycemia with 95% confidence intervals. Here, *p*-value < 0.05 is considered significant. Significant variables were entered into multivariate logistical regression. Abbreviations are as follows: GA—gestational age; SGA—small for gestational age; LGA—large for gestational age; CRIB—clinical risk index for babies; NEC—necrotizing enterocolitis; RDS—respiratory distress syndrome.

**Table 3 medicina-58-01295-t003:** Univariate and multivariate logistic regression for the prediction of hyperglycemia based on perinatal characteristics of preterm infants, mode of delivery, maternal data, and complications.

Variable	Univariate Analysis					Multivariate Analysis					
	Beta	*p*-Value	OR	95% CI	*p*-Value	Beta	*p*-Value	OR		95% CI	*p*-Value
Sex (female)	−0.083	0.877	0.92	0.32–2.64	0.654	Excluded					
GA (weeks)	−0.639	0.032	0.53	0.29–0.94	0.042	−0.059	<0.001	0.55		0.44–0.69	0.002
BW < 1000 g	−0.247	0.765	0.78	0.15–3.94	0.698	Eliminated					
Vaginal delivery	−0.468	0.457	0.63	0.18–2.15	0.598	Eliminated					
Apgar score at 1 min	0.079	0.776	1.08	0.63–1.87	0.453	Eliminated					
Apgar score at 5 min	−0.194	0.592	0.82	0.40–1.68	0.651	Eliminated					
CRIB II	0.122	0.506	1.13	0.79–1.62	0.462	Eliminated					
pH	−13.14	0.047	0.00	0.00–0.87	0.045	Eliminated					
BE	0.123	0.087	1.13	0.98–1.30	0.212	Eliminated					
pCO_2_	−0.480	0.153	0.62	0.32–1.19	0.421	Eliminated					
Antenatal glucocorticoids	3.027	0.078	20.63	0.71–596	0.892	Eliminated					
Pregnancy-induced hypertension	0.364	0.604	1.44	0.36–5.69	0.653	Eliminated					
Maternal diabetes	−1.118	0.197	0.33	0.06–1.78	0.798	Eliminated					
Sepsis	1.713	0.017	5.55	1.35–22.8	0.001	1.962	0.001	7.08		2.2–22.9	0.001
NEC	−1.200	0.270	0.30	0.04–2.54	0.614	Eliminated					
RDS	1.918	<0.001	6.81	3.42–13.6	<0.001	1.565	<0.001		4.78	1.89–2.1	<0.001
Mechanical ventilation	1.157	0.050	3.18	1.01–10.37	0.048	1.080	0.042		2.95	1.03–8.4	0.032

Data are presented as odds ratios for developing hypoglycemia with 95% confidence intervals. Here, *p*-value < 0.05 is considered significant. Significant variables were entered into multivariate logistical regression. Abbreviations are as follows: GA—gestational age; BW—birth weight; CRIB—clinical risk index for babies; NEC—necrotizing enterocolitis; RDS—respiratory distress syndrome.

## Data Availability

The data presented in this study are available on request from the corresponding author.

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
