# Peer review of "Incidence and Risk Factors for Glucose Disturbances in Premature Infants"

_medicina, 2022, doi:10.3390/medicina58091295_

Round 1
Reviewer 1 Report
Summary
This manuscripts reports a prospective examination of glucose levels and correlates of abnormal levels in a large number of premature newborns. The authors measured glucose at least 4 times in the first day of life and at least daily thereafter for a week. They found that abnormal glucose levels are very frequent in this group and that the likelihood of an abnormality was associated with sex, gestational age and other factors.
Strengths
The fact that the study was done prospectively substantially reduces the chance of bias. The definitions of normoglycemia, hypoglycemia, hyperglycemia and unstable glucose are clear and precise. The data analysis was done well and the presentation is clear. The Discussion is relevant to the data.
Areas for Improvement
I assume that IRB approval required getting consent, which would make it surprising that all 466 newborns (lines 73-76) admitted to the NICU and meeting criteria were included. If there were some whose parent(s) did not consent, that should be mentioned or the wording about the included newborns should be made more accurate.
Some information about glucose infusion rate (GIR) is unclear. The statement, “All infants had a glucose infusion rate (GIR) of 4.5 mg/kg/min on the first day of life…” seems to be contradicted by lines 122-123. “The first-line approach for hyperglycemia was to reduce the rate of GIR [sic] by 1-2 mg/kg/min. The minimal GIR could not be lower than 4 mg/kg/min.” (“rate of GIR” is redundant.)
The second half of the paragraph on page 4, lines 153-158 is confusing. It is not clear if “51 of them developed hypoglycemia” includes some who only had a single low glucose or only those who met the definition to be in the hypoglycemic group. Likewise, who are the 105 newborns referred to in “Symptoms of hypoglycemia were noticed in 7/105 infants”? The sentence following that, “Recurrent hypoglycemia was recorded in 6 infants” should probably say, “Recurrent hypoglycemic glucose levels were recorded in 6 infants” because “hypoglycemia” is defined elsewhere and refers to a combination of results.
In paragraphs 3.2 and 3.3 I do not think that citing glucose levels as predictors of hypoglycemia or hyperglycemia is meaningful. I think lines 174-179 and 206-212 should be removed along with tables 3 and 5. Similarly, “and blood glucose level >6.5 mmol/L in the 6th hour of life and in the second day of life” should be removed from line 213.
I do not understand why in Table 2, “LGA” and “Antenatal glucocorticoids” do not have statistics.
Minor Points
A sentence about IRB approval should be included in the Methods. Many readers may not know to find it at the end of the paper.
Expressing glucose in both Standard International Units and “conventional” units is great, but should be maintained throughout the manuscript.
The repeat of the glycemia definitions in lines 110-114 is unnecessary and distracting.
In Figure 1 there is a horizontal line below “28 ≤ 31 weeks” that probably belongs further down.
In Figure 1, “28 ≤ 31 weeks” should be “28 to ≤ 31 weeks”. Likewise for the line below it.
In Tables 2 and 4 were the insignificant factors “Excluded” from consideration or were they “Eliminated” by the regression computation?
In line 216 “sensitivity and specificity” cannot be consider “lower” in the context of a Receiver Operating Characteristics curve. Area under the curve, as stated there, is the appropriate measure.
In paragraph 3.4 line 233 it is not clear whether “is associated with” means there was a significant association.
Typo: line 249: “alike” should be “like”.
I do not think the paragraph in lines 302-310 is relevant to the study and think it should be removed.
Author Response
Dear sir,
thank you for your suggestions. We accepted all your suggestions.

Reviewer 2 Report
This manuscript addresses a question we do not have a conclusive report on. This has the potential to be impactful as previous studies along the same line have had much smaller sample sizes compared to yours.
I had a few questions I was hoping you could clarify:
- a) #77-#86, the definition of hypoglycemia as <45mg/dL - is this only for the first day, first 48 hours, or the whole first week that the babies were studied?
b) Was any distinction made for babies on IV fluids versus formula-fed and purely breastfed babies?
c) If you could clarify the source - a) GIR when hypoglycemia detected in > 34-week babies
b) GIR when hyperglycemia detected n < 28-week babies - #103: GIR 4.5mg/kg/min for all babies or was there a range
- #116: "GIRs were increased to reach the desired levels of blood glucose" - what were the maximum and minimum ranges
- #51: "excessive glucose intake" — may need to be reworded, possibly ‘administration’, since this is iatrogenic and not something the baby decides on their own
- #51-52: administration of "some drugs" that result in hyperglycemia needs to be clarified
- #52: “usually transient” and “in all infants” - seem to be generalizations and either need references or need to be reworded
- #70-76: excluding babies that died from the study prevents determining if glucose imbalance had a role or was a predictor for mortality, exclusion of babies with chromosomal abnormalities prevents the study of glucose abnormalities in those preterm babies. “Severe congenital anomalies” needs to be clarified since 15/466 ~ 3% of total and 15/21 ~71% of excluded babies were excluded for this reason
- #155: definition of symptoms- what kind of symptoms were noted, and what were the corresponding levels for those symptoms? Were similar symptoms noted in normoglycemic babies as well - such as concern for neonatal encephalopathy, sepsis, drug withdrawal, etc
- In the table: Antenatal corticosteroids are also essential to stratify by gestational age since a 36-weeker does not pose the same risks as a 26-weeker, and some Obstetricians may choose not to administer antenatal steroids in the 34-36 week group.
- For babies with persistent hypoglycemia, was critical sample testing performed? To look for metabolic abnormalities, transient or congenital hyperinsulinism, etc.
- Were treatment modalities other than increasing GIR or administering insulin utilized?
- A thorough literature review and re-writing of the manuscript are highly recommended
Author Response
Dear sir,
thank you for your suggestions. We accepted all your suggestion.
Kind regards,
Kristina

Round 2
Reviewer 1 Report
Summary
This manuscripts reports a prospective examination of glucose levels and correlates of abnormal levels in a large number of premature newborns. The authors measured glucose at least 4 times in the first day of life and at least daily thereafter for a week. They found that abnormal glucose levels are very frequent in this group and that the likelihood of an abnormality was associated with sex, gestational age and other factors.
Strengths
The fact that the study was done prospectively substantially reduces the chance of bias. The definitions of normoglycemia, hypoglycemia, hyperglycemia and unstable glucose are clear and precise. The data analysis was done well and the presentation is clear. The Discussion is relevant to the data.
Areas for Improvement
The authors removed the “conventional” units (e.g. mg/dL). They should be restored (in parentheses) as many readers are not yet accustomed to Standard International Units.
Otherwise, all of my concerns have been addressed.
Author Response
Dear sir,
thank you for your suggestion.
We restored in parentheses the "conventional" units - mg/dL in the text (lines 84-92, 175,179,181,185,188, 191,275,277,422).
Kind regards,
Kristina Lah Tomulić